# Cellular Control of Protein Turnover via the Modification of the Amino Terminus

**DOI:** 10.3390/ijms22073545

**Published:** 2021-03-29

**Authors:** Nikola Winter, Maria Novatchkova, Andreas Bachmair

**Affiliations:** 1Max Perutz Labs, Department of Biochemistry and Cell Biology, University of Vienna, A-1030 Vienna, Austria; nikola.winter@univie.ac.at; 2Vienna BioCenter, Research Institute of Molecular Pathology, A-1030 Vienna, Austria; Maria.Novatchkova@imp.ac.at; 3Vienna BioCenter, Institute of Molecular Biotechnology, A-1030 Vienna, Austria

**Keywords:** N-degron, ubiquitin proteasome pathway, autophagy, protein turnover, proteolytic processing, endoprotease

## Abstract

The first amino acid of a protein has an important influence on its metabolic stability. A number of ubiquitin ligases contain binding domains for different amino-terminal residues of their substrates, also known as N-degrons, thereby mediating turnover. This review summarizes, in an exemplary way, both older and more recent findings that unveil how destabilizing amino termini are generated. In most cases, a step of proteolytic cleavage is involved. Among the over 500 proteases encoded in the genome of higher eukaryotes, only a few are known to contribute to the generation of N-degrons. It can, therefore, be expected that many processing paths remain to be discovered.

## 1. Introduction 

N-degrons are degradation signals that reside in the amino terminus of proteins [1,2,3,4,5]. Typically, the amino-terminal residue of N-degrons is essential for their recognition, but other residues in the amino-terminal region modulate the strength of the N-degron. Most N-degrons do not contain the start methionine. How these N-degrons are generated is still unclear in many cases. Considering that only a minor fraction of the over 500 proteases encoded by the average higher eukaryotic genome has been studied extensively, many proteolytic processing and maturation steps that generate N-degrons remain to be characterized. Fortunately, methods to analyze the amino-terminal structure of proteins have been developed and are being refined [6], so that substrates of endoproteolytic cleavage can be identified with increasing efficiency. Here, we review some of the existing data about how N-degrons are generated with an emphasis on plants. Since most knowledge in this respect has been accumulated in animal and fungal models, we will frequently refer to findings from these latter organisms as examples of what can be expected for the plant kingdom. N-degrons with acetylated or formylated amino termini are only briefly addressed at the end. Table 1 summarizes the amino termini we discuss regarding generation, further processing, and the expected impact on protein turnover. 

N-degron recognition usually occurs at dedicated ubiquitin ligases that, with the help of ubiquitin-activating enzymes and ubiquitin-conjugating enzymes, covalently link a ubiquitin chain to the N-degron-carrying substrate. This step is followed by degradation through a large multimeric protease called the 26S proteasome. The process can be modulated by soluble ubiquitin chain-binding receptors. As described in Section 3, autophagy receptors have recently entered the stage, so an autophagy route for the turnover of proteins with certain N-degrons is also possible.

We deduced the amino acid frequencies of the first thirty residues in Arabidopsis thaliana proteins using representative gene models from Araport11 (Figure 1A). Regarding the second residue of open reading frames (ORFs), Table 2 lists the absolute numbers of proteins and relative frequency, which usually deviates from the general abundance of a particular amino acid. The subset of proteins with predicted cytoplasmic or nuclear localization (i.e., proteins with no amino-terminal targeting sequence; Figure 1B) differs from the complete protein set in the amino acid composition of the amino-terminal region. This subset of proteins is of particular interest because enzymes for N-degron-dependent protein turnover reside in the cytoplasm and the nucleus.

As shown in the graphic summary of Figure 1, Leu and Ser dominate positions 3 to 23 when all proteins are considered. This is consistent with the fact that Leu and Ser are the most frequent amino acids in proteins (with ca. 9.5 and 9.1%, respectively; Table 2). In cytoplasmic/nuclear proteins, however, Leu is much less abundant. Amino acids with polar, charged, or small side chains dominate instead. The sequence following residue 23 is more similar between the two protein sets, and Leu is the second most abundant amino acid in both. One may argue that this section is already part of compactly folded domains, so the differences between organelle-targeted and cytoplasmic proteins disappear. Conversely, amino termini of up to 23 residues may be frequently exposed, making them amenable to processing events. In cases where this region is already part of a compact fold, processing has to be initiated on the nascent polypeptide. In the following, we first discuss the cleavage of amino-terminal methionine (Met) as a processing event followed by a number of other proteases with known roles in the generation of N-degrons.

## 2. Methionine Aminopeptidase as a Processing Enzyme towards Protein Destabilization

Methionine aminopeptidases (Met APs) are among the first enzymes to reach nascent amino termini at the ribosome. Several isoforms exist and are conserved among eukaryotes. Although isoforms may have distinct substrate preferences [7,8,9], they share the established cleavage specificity that the amino-terminal Met will be removed if the second residue has a small side chain, but it will not if the second residue has a large side chain.

Proteins with Cys as a second residue (small side chain) are processed in vivo by Met cleavage. A prominent group of plant regulatory proteins belong to the set: ERF-VII class transcription factors, ZPR2, and VRN2 [10,11,12,13]. After a Met AP removes the start methionine, there is another set of enzymes that uses amino-terminal Cys as substrate—the plant cysteine oxidases (PCOs [14]). These enzymes use molecular oxygen to convert Cys into cysteic acid. Once a side chain with negative charge is generated, enzymes of the Arg tRNA protein transferase (ATE) class add an Arg residue onto the modified amino terminus. ATE requires an acidic residue on substrate amino termini—i.e., its activity depends on the oxidation of Cys. The Arg-Cys(ox) terminus is recognized by ubiquitin ligases with a UBR domain [15]. All the characterized UBR domains bind amino termini with basic non-acetylated residues, such as Arg. If binding can be followed by ubiquitylation, which requires a spatially accessible internal Lys residue, the result is the turnover of the substrate by the ubiquitin proteasome system (UPS).

The Cys oxidation pathway has one important physiological input: PCOs have a relatively low affinity for molecular oxygen. This property converts them into oxygen sensors [16]. In tissues with low oxygen pressure, or under conditions of low oxygen—e.g., upon root submergence, amino-terminal Cys is not oxidized. As a consequence, the ensuing proteins remain metabolically stable [10,11]. Interestingly, a functional homolog of PCOs was recently described in humans [17], and this latter enzyme also confers oxygen dependence on substrate amino-terminal Cys oxidation. In plants, as well as in animals, there is another messenger input: the absence of nitric monoxide (NO) also prevents Met-Cys protein turnover [18,19]. The gaseous plant hormone ethylene, in turn, impacts NO levels in plants, allowing the modulation of the pathway by additional physiological and developmental inputs [20].

Amino-terminal Pro can be generated by Met cleavage via Met APs from proteins that start with Met-Pro. In budding yeast, a pathway was discovered that quickly removes several gluconeogenesis enzymes upon the addition of glucose to the medium [21]. The substrate recognition subunit (GID4) of the responsible multi-subunit GID ubiquitin ligase has a binding pocket for amino-terminal Pro. Ligase binding depends on additional residues in the amino-terminal region of potential substrates (amino acids 1–7 of yeast fructose bisphosphatase can act as transferrable N-degron). The ligase is conserved in animals and in plants (putative GID4 homolog of Arabidopsis: At2g37680 according to DIOPT 8.0 [22]). Interestingly, the gluconeogenetic enzymes that led to the discovery of the pathway in *S. cerevisiae* do not start with Pro in most other organisms, including the distantly related fungus *K. lactis*, but the preference of the ligase for amino-terminal Pro is conserved [23]. This implies that additional, possibly conserved, substrates remain to be discovered.

N-terminal Gly can also be generated via Met AP cleavage. A turnover pathway was recently described that may serve as backup to the N-terminal myristoylation of certain Gly amino termini. In case that linkage of the fatty acyl group to such amino termini fails, which would prevent membrane targeting, the protein is removed from the cell [24].

In insects as well as in mammals, a set of polypeptides with pro-apoptotic function has a conserved amino-terminal motif starting with Ala [25]. This motif can act as N-degron. Whereas the insect proteins are cytoplasmic proteins and processed via Met-AP, the mammalian proteins reside in mitochondria, and their N-degrons are generated by mitochondrial processing proteases. The activity and turnover route of these polypeptides are functionally linked to caspase activation and are discussed below (Section 4).

## 3. Processing by Met Aminopeptidases: Exceptions from the Rule

As indicated in Table 1, amino termini with small side chains are not the most prominent N-degrons, and no such case is known in plants so far. Of note, however, Knop and coworkers made a systematic analysis of amino-terminal processing in budding yeast [26]. They found that, if Met is followed by Asn, cleavage can occur by Met aminopeptidase. Similarly, Hwang and coworkers [27] identified Met AP-generated substrates resulting from the Met cleavage of Met-Asn and Met-Gln amino termini. It is likely that cleavage before Asn or Gln depends on the sequence context and is blocked by Met acetylation [26], so that only a fraction of the Met-Asn proteome loses the initiator Met. It is currently unclear whether this cleavage affects a potential substrate quantitatively, or whether only a fraction of a particular substrate gets cleaved. While no example from plants has been published so far, there is a general belief that Met AP specificities are conserved across kingdoms, and enzymes that convert Asn into Asp or Gln into Glu, as well as Arg transferase enzymes that add Arg to an acidic amino-terminal residue, are present in plants. Substrates for this pathway in plants may include proteins linked to biotic defense because Asn deamidase mutants display alterations in pathogen response [28].

## 4. Cleavage of Signal Peptides Can Be Followed by Escape of Proteins from the ER

Proteases that cleave off endoplasmic reticulum (ER) signal sequences frequently expose amino-terminal residues with large side chains. However, because N-degron binding ubiquitin ligases reside in the cytoplasm, this maturation step has no impact on protein stability. The ER can retro-translocate un- or misfolded proteins into the cytoplasm by a pathway called ER-associated degradation (ERAD [29,30]). Retro-translocated proteins are decorated with ubiquitin at the exit channel of retro-translocation (a SEC61 containing membrane protein complex) by dedicated membrane-associated ligases, so amino-terminal recognition may only play a minor role, or no role at all, in marking these proteins with ubiquitin.

A recent finding in mammalian cells, however, generates another link to N-degron recognition in the cytoplasm. Certain forms of stress result in transfer of the (normally) ER-resident chaperone immunoglobulin binding protein (BiP) into the cytoplasm. The protein is apparently functional (implying escape/extrusion from the ER in folded state), but has an amino-terminal Glu residue, emanating from the ER-resident signal peptide cleavage [31]. It is currently unclear how BiP and other proteins escape from the ER, but the exit path for unfolded proteins of ERAD is apparently not involved [32]. Once in the cytoplasm, BiP is arginylated at its amino terminus by Arg transferase ATE. While this amino-terminal processing generates a degron for a UBR domain containing ubiquitin ligases, there is a previously unexpected competition to this process. Two prominent autophagy receptors, p62 and NBR1, have so-called ZZ domains with a high affinity for amino-terminal Arg [33]. One may speculate that under cellular stress, the chaperone BiP is recruited from the ER to bring (cytoplasmic or even ER-generated) misfolded clients. Thereafter, by the binding of BiP to ZZ domains of autophagy receptors, the clients are channeled into autophagy. Interestingly, the occupation of the ZZ domain of p62 promotes its association into larger p62 complexes, which are known to trigger autophagic vesicle formation. Occupancy of these sites by cytoplasmic N-degron proteins can thereby induce ER-phagy [34]. One may speculate that induction of autophagy by accumulation of N-degron substrates, as may occur in the case of suboptimal turnover via the UPS, is not restricted to ER-phagy. While the relevance of the ER protein-initiated turnover route for plants remains to be demonstrated, it should be emphasized that the plant functional homologs of p62 and NBR1, called AtNBR1 in Arabidopsis [35,36] and Joka2 in tobacco [37], contain a ZZ domain and are thus potential receptors for N-degrons. Plant signal peptidases can generate amino-terminal residues that are destabilizing when exposed in the cytoplasm (for instance, Arabidosis BiP2 (identifier At5g42020) is processed in the ER to start with residue Lys 28 according to the prediction, SignalP-5.0 http://www.cbs.dtu.dk/services/SignalP/ (accessed on 26 February 2021) [38], and experimental data, Plant PTM Viewer https://www.psb.ugent.be/webtools/ptm-viewer/ (accessed on 10 March 2021) [39]).

## 5. Caspases, Metacaspases, and Other Endoproteases Related to Cell Death Programs

In mammals and insects, the cell death program apoptosis is, in several ways, interdigitated with N-degron-based turnover. The central elements of the process are proteases called caspases. Their activity in relation to a broad range of substrates leads to cell death. Caspases require proteolytic processing for activation. As the inactive precursors reside in the cytoplasm, several layers of activity containment exist and ensure that cell death is not initiated inappropriately. In particular, the ubiquitin ligases of the inhibitor of apoptosis (IAP) class can associate with caspases to accomplish their degradation. However, at later stages of caspase activation, caspases cleave IAPs to generate N-degrons [40], resulting in IAP degradation. This activity removes one of the breaks on the cell death program [41]. Another layer of regulation is exerted on IAPs. A number of polypeptides of Drosophila (reaper, grim, and hid, in particular) and of mammals (called Smac and Diablo) have a conserved N-terminal motif starting with Ala [25] that binds to so-called Baculovirus IAP Repeat (BIR) domains. IAPs contain BIR domains, and the binding of the mentioned polypeptides prevents association with (and thereby the inactivation of) caspases. It is possible that these polypeptides are eventually ubiquitylated by IAPs and thereby channeled into degradation, but this latter process is probably slow, so that prevention of caspase binding is the dominating effect. As a consequence, grim and related factors are pro-apoptotic elements. It is shown in Drosophila that another BIR domain containing protein called Bruce binds the Ala N-degron and initiates turnover of grim, reaper, and related pro-apoptotic proteins [42]. Bruce is an unusual ubiquitin ligase because it does not contain a RING domain, but a ubiquitin-conjugating enzyme domain instead.

Caspases contain a Cys active site and generally cleave at the carboxyl-terminal end of internal Asp residues. Depending on the family member, substrates are cleaved at the carboxyl end of sequence motifs such as DEVD or LEHD (single letter code [43]). The newly generated amino terminus of the C-terminal fragment (P1’ position; the peptide bond between the so-called P1 and P1’ positions is cleaved, resulting in a fragment with C-terminal P1 residue, and another one with N-terminal P1’ residue) is more variable and thus often encompasses a destabilizing residue. For this reason, these proteases increase the cellular load of N-degron-containing proteins. There is evidence that the UPS, via N-degrons, removes critical caspase substrates at early stages of caspase activation and thereby antagonizes cell death [44,45]. However, it is currently not known whether N-degron-dependent turnover plays a role during the later stages of programmed cell death processes. This is because, in mammals, the proteasome is also a substrate for caspase cleavage, and this leads to proteasome inactivation [46]. Thus, although the amount of N-degron bearing proteins further increases with ongoing apoptosis, other routes of protein degradation may take over for cell clearance.

In plants, close homologs of caspases do not exist, but metacaspases have a related role [47,48]. Plant metacaspases do not have the Asp specificity for the P1 site, but rather cleave C-terminal to Arg or Lys. Tsiatsiani et al. [49] found, however, a preference for Asp and Glu in the P1´ site, indicating that plant metacaspases also generate N-degrons. Interestingly, some plant proteases with a Ser active site apparently share a preference for Asp or Asn in the P1 position, and may therefore have a cleavage specificity that is more similar to animal caspases. For instance, the vacuolar processing enzyme (VPE) class (also called legumains), recognizes cleavage sites with Asn or Asp in P1 position [50,51]. One critical feature of these enzymes is that they normally reside in the extracellular space (a Ser protease called phytaspase [52]) or in the vacuole (VPEs), so protein fragments generated at the normal locale of these enzymes are sequestered from the cytoplasm with its active N-degron pathway enzymes. However, both types of proteases have been implicated in cell death processes, whereby these enzymes do reach the cytoplasm [52,53]. Finally, the PBA1 subunit of the proteasome, a Thr protease, was also reported to have a caspase-like cleavage specificity in vitro, and its absence impacts on bacterially induced plant cell death [54]. In the intact proteasome, this activity is shielded from the cytoplasm, and PBA1 is produced as an inactive precursor prior to assembly into proteasomes. Therefore, this enzyme also needs to be released into the cytoplasm by a currently unknown mechanism before it can generate N-degron bearing proteins.

Similarly, the plant-specific cell death, senescence, has been linked to endoproteases such as those discussed above. With many indications of varying cellular localization, one may assume that the abundance and/or localization, but not the cleavage specificity, of these proteases depends on the biological context. It is interesting to mention, in this regard, that a mutation in the Arg transferase ATE1, which converts Asp and Glu amino termini into Arg-Asp and Arg-Glu termini, has a delayed senescence phenotype [55]. However, Arg transfer to termini generated by senescence-associated endopeptidases has not been demonstrated so far.

## 6. Bacterial Effectors

A number of bacterial effectors that are injected into the plant cytoplasm via a type three secretion system contain protease activity. Some of them show activity as de-SUMOylating proteases that release SUMO from conjugates (XopD [56]), but many effectors with protease activity may be active against a broader range of plant proteins, often with key roles in plant defense. A well-known example is AvrRpt2. This protease targets an important component of basal immunity, RIN4 [57,58]. Released fragments have the destabilizing neo-N-termini Asn and Asp [59] (see also Table 1). While these RIN fragments are metabolically unstable, they are apparently not stabilized in the *prt6* N-degron ubiquitin ligase mutant. However, AvrRpt2 also cleaves related proteins, such as other members of the NOI domain family to which RIN4 belongs, and some of the resulting fragments accumulate in N-degron turnover mutants such as *prt6* [59]. It is therefore possible that RIN 4 and some of its NOI domain protein relatives contain additional cryptic degradation signals that are exposed by endoproteolytic cleavage.

## 7. Other Specialized Endoproteases

All eukaryotes contain separases, which are necessary to cleave cohesins at the end of mitosis, but they may have other substrates, as well [60]. It has been shown that failure of N-degron-based removal of cohesin fragments after separase cleavage decreases the precision of cell cycle events [61].

In mammals, the calcium-activated proteases, calpains, can also generate N-degrons [62]. Plants have a single copy gene related to calpains—defective kernel 1 (DEK1). DEK1 has an essential function in growth coordination between epidermis and inner leaf cell layers [63]. DEK1 is a fusion of a calpain domain to a transmembrane transporter that may be a touch-sensitive ion channel. DEK1 is functionally coupled to a Ca transporter, but Ca transport activity may reside in a separate polypeptide [64,65]. Activated DEK1 undergoes auto-cleavage, but the released protease may still remain associated with membranes. Unfortunately, the cleavage specificity of DEK1 has not been elucidated yet, and no substrates are known so far.

## 8. No Proteolytic Processing of Polypeptides Emerging from the Ribosome: Can Met Be a Destabilizing Residue If Followed by an Amino Acid with Hydrophobic Side Chain?

The prototypic N-degron ubiquitin ligase, Ubr1 of *S. cerevisiae*, has a binding pocket for hydrophobic amino termini, such as Leu, called ClpS homology domain [66]. Animal homologs have a similar domain called N domain [67]. This domain is apparently not conserved in plant homologs of UBR1 type ubiquitin ligases, and binding proteins for aliphatic hydrophobic residues remain to be discovered. For Ubr1, however, protein substrates binding to the ClpS homology domain have been described that start with Met, followed by a hydrophobic amino acid [68]. Another yeast ubiquitin ligase, Doa10, also seems to be involved in Met-Φ (Φ symbolizes a bulky hydrophobic amino acid) substrate turnover [26]. Because more than ten percent of the Arabidopsis ORFs start with Met-Φ (Table 2), it is highly likely that only a fraction of them are substrates for rapid turnover. The additional selection criteria for this class of substrates remain to be determined. All in all, we should be open to the possibility that the selection process for hydrophobic amino termini as N-degrons in plants may also include a fraction of the Met-Φ proteome.

## 9. Blocking the Alpha-Amino Group Can Generate a Distinct Set of N-Degrons

All the abovementioned amino termini carry a free, positively charged alpha amino group. In contrast, modification by acetylation or by formylation abrogates this positive charge, lowering affinity for the above-mentioned binding pockets on ubiquitin ligases or autophagy substrate adaptors. Nonetheless, N-degrons with a blocked first residue have been discovered, fueling dedicated turnover pathways.

In baker´s yeast, ubiquitin ligases Doa10 and Not4 mediate the degradation of certain proteins dependent on an acetylated first residue, making these ligases key components in the recognition of the specific class of Ac/N-degrons [69]. In plants as in yeast, amino-terminal acetylation does not destabilize the bulk of cytoplasmic proteins [70]. However, this does not rule out the existence of specific plant proteins with Ac/N-degrons because both Doa10 and Not4 [71], in particular, have homologs in plants.

Similarly, proteins in yeast can be modified upon stress or starvation on N-terminal Met residues by formylation, and ubiquitin ligase Psh1 is reported to mediate the removal of formylated proteins from the cytoplasm [72]. Here, again, plants encode proteins related to the recognin (ORTH/VIM proteins [73]), but functional connections to fMet N-degron generation or degradation remain to be demonstrated in plants. The further discussion of these two pathways is beyond the scope of this review.

In conclusion, the elucidation of cleavage specificities of currently uncharacterized proteases would reveal a potential role for protein stability. Even for established enzymes such as Met APs, it would be important to obtain additional, quantitative information regarding cleavage preferences (e.g., in the context of the cleavages described in Section 3).

Furthermore, progress is expected to come from the proteomic identification of protein fragments [6]. Once the occurrence of a certain cleavage pattern has been established, experiments to identify the processing path can be initiated. Alternatively, the over-expression of particular proteases, followed by proteome analysis, is also a promising strategy [49]. As a third route to pathway elucidation, the further characterization of degradation components, particularly the affinities of N-degron binding pockets in ubiquitin ligases and autophagy receptors for certain amino termini, can also help us to fill in the knowledge gaps. 

## Figures and Tables

**Figure 1 ijms-22-03545-f001:**
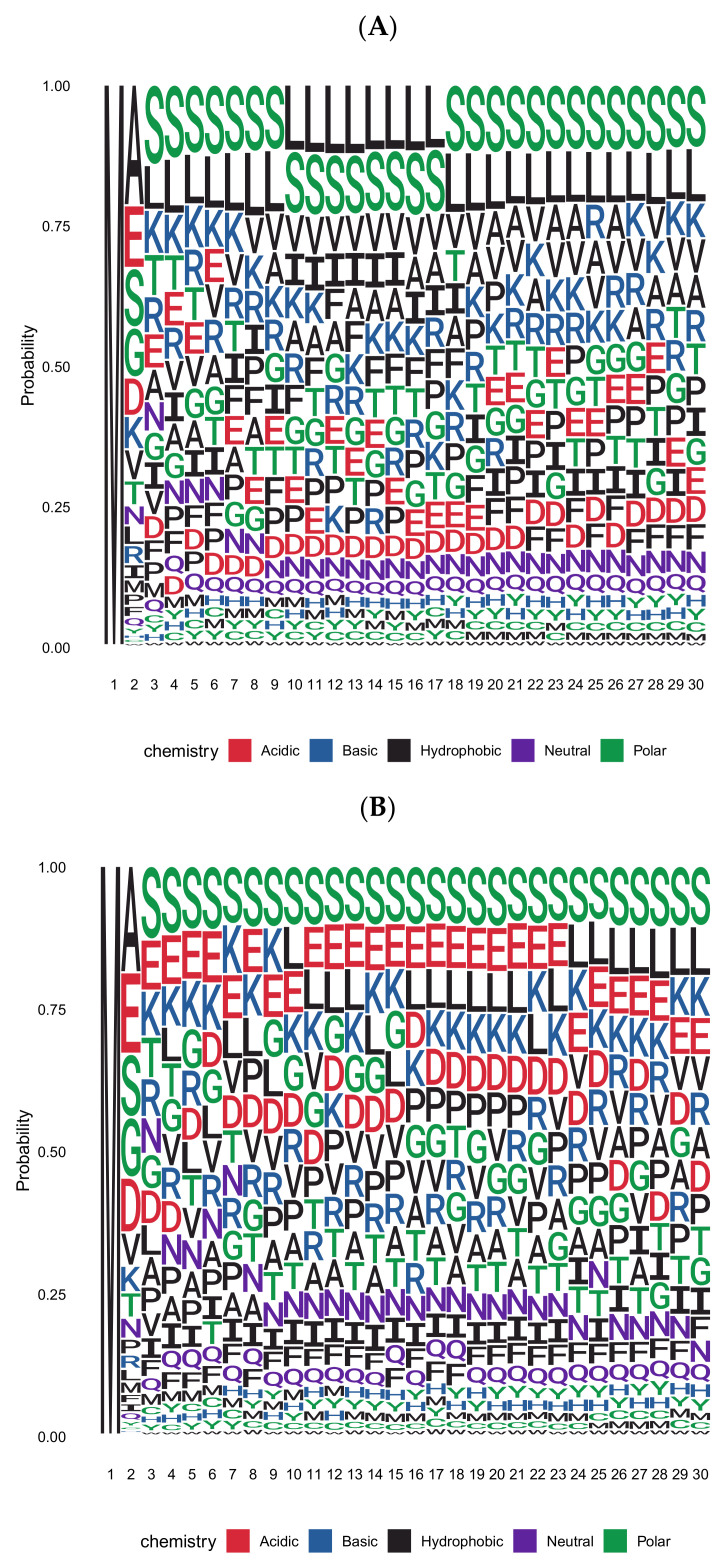
Position-specific amino acid probability in the first thirty residues of Arabidopsis proteins represented as a sequence logo (in single letter code). (**A**) Sequence logo based on the complete set of nuclear-encoded proteins with one representative per gene. (**B**) Sequence logo based on the subset of proteins with predicted localization exclusively in cytoplasmic or nuclear compartments. Predictions were obtained from Araport11.

**Table 1 ijms-22-03545-t001:** Fate of proteins with the indicated amino-terminal residues.

Exposed Residue	Exposing Protease	Predicted Fate
Cys	Met aminopeptidase (Met AP)	Oxidation, arginylation, ubiquitylation, proteasomal degradation of protein
Pro	Met AP	Depending on sequence context: Pro N-degron
Gly	Met AP	Depending on sequence context: Gly N-degron
Ala	Met AP/mitochondrial protease	Depending on sequence context: Ala N-degron
Asn	Met AP (depending on sequence context; blocked by Met acetylation), other endoproteases?	De-amidation, arginylation, ubiquitylation, proteasomal degradation of protein
Gln	Met AP (depending on sequence context; blocked by Met acetylation), Metacaspases, other endoproteases?	De-amidation, arginylation, ubiquitylation, proteasomal degradation of protein
Asp	Caspase-like proteases, other endoproteases	Arginylation, ubiquitylation, proteasomal degradation of protein
Glu	Signal peptidase, other endoproteases?	Arginylation, ubiquitylation, proteasomal degradation of protein
Lys	Metacaspases	Ubiquitylation, proteasomal degradation of protein
Arg	Metacaspases	Ubiquitylation, proteasomal degradation of protein
Met Φ ^1^	No processing	Depending on sequence context: ubiquitylation, proteasomal degradation of protein

^1^ Φ symbolizes an amino acid with hydrophobic side chain.

**Table 2 ijms-22-03545-t002:** Abundance of amino termini in the predicted proteome of Arabidopsis ^1^.

Amino-Terminal Dipeptide	Abundance in the Compete Proteome	Abundance (%)	Abundance in Nuclear/Cytoplasmic Proteins	Abundance (%)	General Frequency of Amino Acid 2 in the Proteome
MA	5891	21.5%	1917	18.7%	6.3%
ME	3059	11.1%	1469	14.3%	6.7%
MS	2898	10.6%	1140	11.1%	9.1%
MG	2441	8.9%	1090	10.6%	6.4%
MD	1874	6.8%	988	9.6%	5.4%
MK	1629	5.9%	479	4.7%	6.4%
MV	1519	5.5%	598	5.8%	6.7%
MT	1216	4.4%	458	4.5%	5.1%
MN	976	3.6%	378	3.7%	4.4%
ML	951	3.5%	225	2.2%	9.5%
MR	931	3.4%	259	2.5%	5.4%
MI	738	2.7%	162	1.6%	5.3%
MM	692	2.5%	225	2.2%	2.5%
MP	601	2.2%	285	2.8%	4.8%
MF	544	2.0%	167	1.6%	4.3%
MQ	509	1.9%	155	1.5%	3.5%
MY	340	1.2%	101	1.0%	2.8%
MH	208	0.8%	51	0.5%	2.3%
MC	205	0.7%	68	0.7%	1.9%
MW	164	0.6%	41	0.4%	1.2%
Sum	27,386		10,256		

^1^ For each gene identifier of Araport11 version 20160703, a single representative gene model was used as provided by Arabidopsis.org (accessed on 26 February 2021); genes from ChrM and ChrC were excluded.

## Data Availability

Data supporting reported results can be found in the quoted references.

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
