# Peer review of "Cellular Control of Protein Turnover via the Modification of the Amino Terminus"

_ijms, 2021, doi:10.3390/ijms22073545_

Round 1

Reviewer 1 Report

This article by Winter et al presents a nice overview of amino-terminus triggered protein degradation, with a focus on known and putative proteolytic events that can promote N-terminal exposure to target substrates for destruction.  To my knowledge, this is a slightly different angle than is usually taken, and so it gives the review a fresh narrative.  Overall, the work is of general interest to the field of plant proteostasis and I believe it covers the breadth of literature.  However, I do have several minor comments (including grammar/spelling corrections) that should be addressed:

Line 22: I think that “degradation signals” sounds better than “turnover signals”

Line 35: should say “termini” not “terminus”

Line 39: should say “ubiquitin activating enzymes

Line 41: should say “called THE 26S proteasome”

Line 53: consider rewording this sentence: “The cytoplasmic / nuclear protein subset has more often an acidic second residue, and less often a hydrophobic or basic residue at position two than the total proteome (Table 2)”

Line 59: It wasn’t initially clear to me what was meant by: “Only after amino acid 24, Leu is the second 59 most abundant residue in both protein sets”.  Perhaps authors could revise this statement.

Line 62 “The differences suggest that amino termini of up to 24 62 residues my be frequently exposed, making them amenable to processing events.” I am not sure the data presented are sufficient enough to support this conclusion?  Also check spelling of “residues” and “may”.

Line 75: “Many of these proteins have the fate of being short-lived”.  I would not say “many of these…” – to date in plants, ERF-VIIs (e.g., Licausi et al and Gibbs et al 2011 Nature), ZPR2 (Weits et al 2019 Nature) and VRN2 (Gibbs et al 2018 Nature Comms) are the only known Cys N-degron pathway targets out of approx. 250 Met-Cys proteins (Gibbs et al 2014 Trends in Plant Science) encoded by the Arabidopsis genome.  Please revise this statement to reflect this.

In fact, it might be worth mentioning these specific substrates (and their associated references) in this or the subsequent paragraph.  

Line 117: “The activity and turnover route of these polypeptides is functionally linked to caspase activation, and is discussed and referenced below (section 4)” I still think that he references should be cited here, even if the details are not described.

Line 142 – remove the word “getting”

Line 257: should say “can also generate”

Line 262: should say separate

Author Response

We want to thank the reviewer for his/her constructive and helpful comments.

We have incorporated all suggestions. Both language corrections, and suggestions to change parts of the text. Regarding the latter, we re-phrased parts of the introduction (lines 45 ff), and of section 2 (second paragraph, starting with line 74). In this part of section 2, additional references were included. Furthermore, the last paragraph of section 2 (lines 114ff) was changed to include a reference (now ref. 25) that was previously only quoted later in the text.

Reviewer 2 Report

In this manuscript, Winter and coworkers reviewed the dominant cellular mechanisms to degrade proteins via N-termini recognitions. Such information for N-degrons was well organized into the canonical Met aminopeptidases, ER-secreted aminopeptidases, the caspase family, and others. The manuscript was well written with the balanced materials of canonical and newly emerged pathways. The manuscript is timely given the increased interest in protein homeostasis. The reviewer recommended publishing this work after addressing the following minor concerns:

(1) In Session 2 – methionine aminopeptidase: while the authors nicely listed and summarized the various classes of aminopeptidases on the basis of their abilities to recognize the N+2,3,4 … sequences, the authors didn’t describe how each of such activities contribute the products in a collective or competitive manner. It will be helpful to semi-quantitatively describe the relative contribution of these activities and their rankings. The current tone sounds like that one pathway is dominant in a specific cellular context.

(2) For the same reason, the relative contribution of the exceptional cases should be presented in a ranked and semi-quantitative manner.  

(3) In the last paragraph, the authors briefly described the effect of Ac-N. However, it is also important to summarize other type N-terminal capping mechanisms including methylation and formylation.

Author Response

We want to thank the reviewer for his/her constructive and helpful comments.

The revewer suggested to include a statement on formyl-Met as a degradation signal. This was achieved by changes in section 9. We also changed the first paragraph of section 2 (lines 69ff) in order to point out that Met-APs are a group of enzymes with different specificities.

Another suggestion of the reviewer was to rank different pathways and activities in a semi-quantitative way. We agree with the reviewer that this would further increase the value of the review. Unfortunately, there are simply not enough data available to achieve this goal. Systematic quantitative analysis has started with proteomic analysis of amino termini and with the work of the Knop group (ref. 26), but for the majority of the mentioned cases, a ranking cannot be based on the available data. We hope that our final statements make it clear that quantitative approaches are on the path to further progress.